# Diffusion of Vanadium Ions in Artificial Saliva and Its Elimination from the Oral Cavity by Pharmacological Compounds Present in Mouthwashes

**DOI:** 10.3390/biom12070947

**Published:** 2022-07-06

**Authors:** Sónia I. G. Fangaia, Ana M. T. D. P. V. Cabral, Pedro M. G. Nicolau, Fernando A. D. R. A. Guerra, M. Melia Rodrigo, Ana C. F. Ribeiro, Artur J. M. Valente, Miguel A. Esteso

**Affiliations:** 1Institute of Implantology and Prosthodontics, CIROS, Faculty of Medicine, University of Coimbra, 3000-075 Coimbra, Portugal; pgnicolau@mail.telepac.pt (P.M.G.N.); fguerra@ci.uc.pt (F.A.D.R.A.G.); 2Faculty of Pharmacy, University of Coimbra, 3000-548 Coimbra, Portugal; acabral@ff.uc.pt; 3Department of Chemistry, CQC-IMS, Institute of Molecular Sciences, University of Coimbra, 3004-535 Coimbra, Portugal; avalente@ci.uc.pt; 4U.D. Química Física, Universidad de Alcalá, 28805 Alcalá de Henares, Spain; mmelia.rodrigo@uah.es (M.M.R.); mangel.esteso@ucavila.es (M.A.E.); 5Universidad Católica de Ávila, 05005 Ávila, Spain

**Keywords:** cyclodextrins, diffusion, hyaluronic acid, ion release, mouthwashes, SDS, Ti-6Al-4V, vanadium

## Abstract

In this study, diffusion coefficients of ammonium vanadate at tracer concentrations in artificial saliva with and without sodium fluoride, at different pH values, were measured using an experimental model based on the Taylor dispersion technique. Ternary mutual diffusion coefficients (*D*_11_, *D*_22_, *D*_12_, and *D*_21_) for four aqueous systems {NH_4_VO_3_ (component 1) + β-cyclodextrin (β-CD) (component 2),} {NH_4_VO_3_ (component 1) + β-cyclodextrin (HP-β-CD) (component 2)}, {NH_4_VO_3_ (component 1) + sodium dodecyl sulphate (SDS) (component 2)} and {NH_4_VO_3_ (component 1) + sodium hyaluronate (NaHy) (component 2)} at 25.00 °C were also measured by using the same technique. These data showed that diffusion of ammonium vanadate was strongly affected in all aqueous media studied. Furthermore, a significant coupled diffusion of this salt and β-CD was observed through the non-zero values of the cross-diffusion coefficients, *D*_12_, allowing us to conclude that there is a strong interaction between these two components. This finding is very promising considering the removal, from the oral cavity, of vanadium resulting from tribocorrosion of Ti-6Al-4V prosthetic devices.

## 1. Introduction

Titanium–aluminum–vanadium alloys are commonly used in the fabrication of orthopedic and dental implants, as well as in dental prosthetic supra-structures, abutments, and healing components, due to their good mechanical properties and biocompatibility [1,2,3].

However, as they are inserted into the oral cavity, they can be subjected to corrosion and wear phenomena, with consequent release of metal ions into the human body [4,5,6]. These released metal ions and particles may have deleterious systemic and local effects; for instance, they are related to the failure of implant-supported oral rehabilitations and thus matter of concern for the scientific community [7,8,9,10,11,12,13,14].

According to Barrak et al. [15], although titanium-based prostheses are considered non-toxic, the same may not be true with particles and ions released from them. Consequently, the potential cytotoxicity of Ti-6Al-4V implant particles should be highlighted, and more investigations on the biological effect of the fine particles or metallic species released are needed. This standpoint is corroborated by Kunrath et al. [12], who considered the long-term behavior of fragments of certain chemical elements in tissues or cells at the molecular level and indicated that this is still not well-understood. Ion release from metallic prosthetic devices is a current issue and the object of many recent studies [11,13,14,15,16].

In this paper, we focus on vanadium; according to Zwolak [17], deleterious effects induced by inorganic vanadium compounds are linked with carcinogenic, immunotoxic, and neurotoxic insults. Furthermore, other studies suggest that vanadium interferes in the hemopoietic system in rats [18] and a significant decrease in the fibroblasts’ cell viability has also been observed [8]. Due to its suspected strong cellular toxicity and tissue accumulation, resulting in severe damage, [19,20] their release from prosthetic devices to the oral cavity is a matter of concern.

In oral hygiene, mouth rinses are used for several clinical purposes such as chemical plaque control, treatment of periodontal diseases, oral lesions, halitosis, prevention of dental caries, bleaching, etc.

Concerning the therapeutic mouth rinses constituents, we can highlight active ingredients such as chlorhexidine [21], triclosan, cetylpyridinium chloride [22,23], hyaluronic acid (HA) [24,25,26], sodium dodecyl sulfate (SDS) [27,28], and cyclodextrins (CDs) [21,29].

The most common natural cyclodextrins are α, β, and γ–cyclodextrins, containing, respectively, six, seven, and eight glucopyranose units [30,31]. The chair conformation of the glucopyranose units, gives cyclodextrins a shape of a truncated cone, with a hydrophilic exterior surface and hydrophobic interior cavity [32]. Due to this particular molecular arrangement, they can trap guest hydrophobic molecules inside their cavity and act as molecular containers [30]. These properties make CDs versatile compounds, which, along with their biocompatibility, allows their application in, e.g., pharmacology [33]. In the context of our study, the fact that β-cyclodextrin (β-CDs) are resistant to hydrolyzation via salivary amylases [34] is of particular importance. Therefore, in this study, we used β-CD and HP-β-CD, a CD modified with a functional group with higher solubility [35].

Hyaluronan, commonly known as hyaluronic acid (HA), has been used in the constitution of mouth rinses due to its anti-inflammatory and antioxidant properties and bacteriostatic effect [36,37]. It is a linear macromolecular mucopolysaccharide that is composed of alternatingly linked two saccharide units of glucuronic acid and N-acetylglucosamine, with good biocompatibility and biodegradability [38,39]. According to Vasvani et al. [40], because of a sufficient electric charge, HA can attract positive ions, having a great therapeutic potential either as a combinative agent with encapsulation of different drugs and biomolecules or in the form of a nanocarrier itself.

As for SDS, it is an anionic surfactant, with a hydrophobic chain length capable of dissolving if SDS concentrations are below critical micelle concentration (CMC) the outer layer of viruses and bacteria, and a hydrophilic head that dissolves in water. It is often used in the constitution of detergents, liquid soaps, and toothpaste; having been used in mouthwashes, it is the subject of many studies due to its potential virucidal activity [27,41].

As for cyclodextrins [32] and hyaluronic acid [38], SDS is also currently used in drug delivery systems, due to their capacity to interact with biomolecules [42].

The objective of this study was to analyze the diffusion behavior of vanadium in artificial saliva with and without fluoride, at different pH values, using an experimental model based on the Taylor dispersion technique [43,44]. Furthermore, having in mind the capacity of hyaluronic acid, SDS, and CDs to interact with biomolecules, we aim to evaluate if those pharmacological agents, present in mouthwashes, can interact with vanadium ions, facilitating its removal from the oral cavity during mouthwash.

## 2. Materials and Methods

### 2.1. Materials

Ammonium vanadate (Riedel-de-Haen, Seelze, Germany, pro-analysis > 97%), sodium fluoride, lactic acid, SDS, NaHy, and β-cyclodextrin (β-CD) and one of its derivatives, HP-β-cyclodextrin (HP-β-CD) were used without further purification (Table 1); after drying, they were stored in a desiccator over silica gel. Artificial saliva was prepared according to the composition indicated in Table 1.

Solutions for the diffusion measurements were prepared using Millipore-Q (Milli-Q^®^ EQ 7000 Ultrapure Water Purification System-Merck Millipore, Darmstadt, Germany) water (specific resistance = 1.82 × 10^5^ Ω m, at 25.00 °C). All solutions were freshly prepared at 25.00 °C before each experiment. The weighing was performed using a Radwag AS 220C2 balance (Precision scale AND, A&D Instruments Ltd., Oxford, UK) with a readability of 10^−5^ g in the lower range.

### 2.2. Measurements of Diffusion Coefficients

#### 2.2.1. Phenomenology of the Diffusion for Different Systems (Binary, Pseudo-Binary, Ternary, and Pseudo-Systems)

The isothermal diffusion is an irreversible phenomenon, resulting from the gradient of chemical potential in the real solution. However, in dilute solutions, this force, responsible for this process, can be quantified by the gradient of the concentration at a constant temperature, and in binary systems (i.e., with two independent components), this parameter may be defined in terms of the concentration gradient by a phenomenological relation, known as Fick’s first law (Equation (1)).
(1) J (solute)=−D∇C
where *D*, *J*, and ∇*C* represent the binary diffusion coefficient, the molar flux, and the gradient in the concentrations of solute, respectively.

Diffusion in a ternary solution (that is, two solutes and water) is described by diffusion equations (Equations (2) and (3)) as follows:(2)$$J_{1}\;\left(\mathrm{solute}\;1\right)=-D_{11}\nabla C_{1}-D_{12}\nabla C_{2}$$
(3)$$J_{2}\;\left(\mathrm{solute}\;2\right)=-D_{21}\nabla C_{1}-D_{22}\nabla C_{2}$$
where *J*_1_ and *J*_2_ are the molar fluxes of component 1 and component 2 driven by the concentration gradients **∇***C*_1_ and **∇***C*_2_ of each solute 1 and solute 2, respectively. Main diffusion coefficients *D*_11_ and *D*_22_ give the flux of each solute driven by its own concentration gradient. Cross-diffusion coefficients *D*_12_ and *D*_21_ give the coupled flux of each solute driven by a concentration gradient in the other solute. A positive *D*_ab_ cross-coefficient (a ≠ b) indicates co-current coupled transport of solute “a” from regions of higher to lower concentrations of solute “b”. On the other hand, a negative *D*_ab_ coefficient indicates counter-current-coupled transport of solute “a” from regions of lower to higher concentration of solute “b”.

In the present study, the diffusion coefficient of NH_4_VO_3_ in water was measured. The diffusion of this electrolyte, described by Fick’s law (Equation (1)) with a single diffusion coefficient *D*, is a weighted average of the diffusion coefficients of the ionic species and its counter-ions. The anions and cations of this electrolyte diffuse at the same speed to maintain electroneutrality along the diffusion path.

In addition, we also measured the diffusion of this salt in artificial saliva, with and without different components, and at different pH values. In these situations, the systems are actually multicomponent systems; that is, there are more than two independent components. However, from experimental conditions (details are described below), we may consider these systems as pseudo-binary ones and, consequently, take the measured parameter (*D*_11_) as binary diffusion coefficients, *D,* at tracer concentrations (Equation (4)) as follows:*J*_1_(NH_4_VO_3_, tracer) = −*D*_11_∇*C*_1_(4)

Although one should rigorously distinguish parameters *D* (Equation (1)) and *D*_11_ (Equation (4)) as obtained for binary (NH_4_VO_3_/H_2_O) and pseudo-binary systems (NH_4_VO_3_/artificial saliva(water), from now on, for simplicity, only the terminology diffusion coefficient of NH_4_VO_3_ in these media were used in our approach.

In the last part of this study, the effect of two cyclodextrins (β-CD and HP-β-CD)—sodium dodecyl sulfate (SDS) and sodium hyaluronate (NaHy)—on the diffusion of aqueous ammonium monovanadate was also investigated by measuring ternary mutual diffusion coefficients for aqueous NH_4_VO_3_ (1) + β-CD or (HP-β-CD, component 2) solutions. The host ability of CDs was compared with that of sodium dodecyl sulfate and sodium hyaluronate. In this case, however, it is worth noting that measured parameters (*D*_ij_) are designated as apparent diffusion coefficient values once these ternary systems are truly quaternary systems. Nevertheless, these data prove to be useful for qualitatively understanding the main features of these different systems.

#### 2.2.2. A Summary Description of These Measurements by Using the Taylor Technique

The Taylor dispersion method, which has been used to obtain mutual diffusion coefficients for a wide variety of aqueous binary and ternary systems, is described in great detail in some studies [47,48]. Basically, it consists of the dispersion of small amounts of solution injected into laminar carrier streams of water or solution of different compositions, flowing through a long capillary tube with a length and radius of 3.2799 (±0.0001) × 10^4^ mm and 0.5570 (±0.00003) mm, respectively [49,50].

At the start of each run, a 6-port Teflon injection valve (Rheodyne, model 5020, Sigma-Aldrich, Darmstadt, Germany) is used to introduce 0.063 mL of solution into the laminar carrier stream of a slightly different composition. Using a metering pump (model Minipuls 3, Gilson, Middleton, WI, USA), a flow rate of 0.17 mL min^−1^ is maintained to give retention times of about 1.1 × 10^4^ s. The injection valve and the dispersion tube are kept at 25.00 (±0.01) °C in an air thermostat.

Dispersion of the injected samples is monitored using a differential refractometer (model 2410, Waters, Milford, MA, USA) at the outlet of the dispersion tube. Detector voltages, *V*(*t*), are measured at accurately 5 s intervals with a digital voltmeter (Agilent 34401 A, Santa Clara, CA, USA) with an IEEE interface. Binary diffusion coefficients of NH_4_VO_3_ in water are evaluated by fitting the dispersion equation
*V*(*t*) = *V*_0_ + *V*_1_*t* + *V*_max_ (*t*_R_/*t*)^1/2^ exp[−12*D*(*t* − *t*_R_)^2^/*r*^2^*t*](5)
to the detected voltages. The additional fitting parameters are the mean sample retention time *t*_R_, peak height *V*_max_, baseline voltage *V*_0_, and baseline slope *V*_1_. In the present study, the binary diffusion coefficient of NH_4_VO_3_ at infinitesimal concentration was obtained (Figure 1), and the dispersion profiles were prepared by injecting different solutions of NH_4_VO_3_ at different concentrations (i.e., 0.001, 0.002, 0.005, 0.008 and 0.010 mol dm^−^^3^) into water. As this salt is only present in the injected solutions, once the equation parameters above were obtained at tracer concentrations, limiting diffusion coefficients could be measured at tracer concentrations.

Relative to the diffusion coefficients of NH_4_VO_3_ in artificial saliva (at different pH, without and with NaF component; Table 2), these profiles were obtained by injecting some amount of this artificial saliva with NH_4_VO_3_ 0.001 mol dm^−3^ into carrier streams of the same artificial saliva; that is, the flow and injected solutions of compositions are *c*_1_ = 0 and *c*_2_ = *c*_2_, and *c*_1_ = Δ*c* and *c*_2_ = *c*, respectively, the detector signal resembles a single normal distribution with variance 2*t*_R_/24*D*_11_, and there are no two overlapping normal distributions. Thus, we may consider the systems pseudo-binary as an approach and, consequently, take the measured parameters as the tracer diffusion coefficients of NH_4_VO_3_ in artificial saliva. The respective diffusion coefficients were evaluated by fitting the dispersion equation (Equation (4)).

Extensions of the Taylor dispersion technique were used to measure mutual diffusion coefficients (*D*_ij_) for two ternary aqueous solutions (that is, NH_4_VO_3_ plus β-CD, and NH_4_VO_3_ plus HP-β-CD). These *D*_ij_ coefficients, defined by Equations (2) and (3), were evaluated by fitting the ternary dispersion equation (Equation (6)) to two or more replicate pairs of peaks for each carrier stream.
(6)$$V\left(t\right)=V_{0}+V_{1}t+V_{\max}{\left(t_{R}/t\right)}^{1/2}\left[W_{1}\exp\left(-\frac{12D_{1}{\left(t-t_{R}\right)}^{2}}{r^{2}t}\right)+(1-W_{1})\exp\left(-\frac{12D_{2}{\left(t-t_{R}\right)}^{2}}{r^{2}t}\right)\right]$$

Two pairs of refractive index profiles, *D*_1_ and *D*_2_, are the eigenvalues of the matrix of the ternary *D*_ab_ coefficients. In these particular experiments, small volumes of ΔV of solution, of composition *C*_1_ + **Δ***C*_1_ and *C*_2_ + **Δ***C*_2_ are injected into carrier solutions of composition, *C*_1_ and *C*_2_ and, at time *t* = 0. More details about the experimental procedure involved in obtaining these parameters may be found in the literature [51].

The Taylor technique was also used to measure pseudo-ternary diffusion coefficients in two aqueous systems, that is, {NH_4_VO_3_ (1) + SDS (2)} and {NH_4_VO_3_ (1) + NaHy (2)} by injecting sample of {NH_4_VO_3_ + SDS (or NaHy)} solutions of composition (*C*_1_ + **Δ***C*_1_), (*C*_2_ + **Δ***C*_2_) into carrier streams of composition *C*_1_ and *C*_2_.

In these particular cases, the system’s target is quaternary and not a ternary system, and thus, coupled diffusion produces apparent ternary dispersion profiles, resulting in apparent diffusion coefficient values or pseudo-ternary diffusion coefficients [52].

However, in practice, measurement of the nine quaternary *D*_ik_ coefficients is very difficult, particularly for these systems. On the other hand, no theory is yet available to reliably predict these diffusion coefficients with the accuracy demanded by the technology and scientific community. Thus, to try to solve these problems, we measured pseudo-ternary diffusion coefficients (*D*_ik*a*_) for NH_4_VO_3_ (*C*_1_) + SDS (or NaHy, *C*_2_) solutions by assuming that the flux of NaVO_3_ (*C*_3_), the third electrolyte, is negligible. Support for this approximation was given in other similar studies [52].

### 2.3. pH Measurements

The pH measurements of solutions were carried out with a Radiometer pH meter PHM 240 (Radiometer analytical SAS, Villeurbanne, France) with an Ingold U457-K7pH conjugated electrode. The electrode was immediately calibrated before each experimental set of fresh solutions using IUPAC-recommended pH 4, 7, and 10 buffers. From pH meter calibration, a zero pH of (6.400 ± 0.030) and a sensitivity higher than 98.7% were obtained. To perform these measurements at pH 2.3 and 7.0, the intended values of the pH were adjusted by the addition of lactic acid. All solutions were freshly prepared at 25.00 °C and degassed by sonication for about 60 min before each experiment.

## 3. Results

Figure 1 shows the dependence of NH_4_VO_3_ diffusion coefficients on pH, in different media, that is, water and artificial saliva with and without other components (lactic acid, sodium fluoride, and the mixture of both), at different pH values. These parameters were measured at tracer concentrations because, in real situations, the concentration of these ions resulting from the corrosion of the dental alloys is very small [53]. These values were the average ones obtained from at least four independent experiments (reproducibility better than 2%).

**Figure 1 biomolecules-12-00947-f001:**
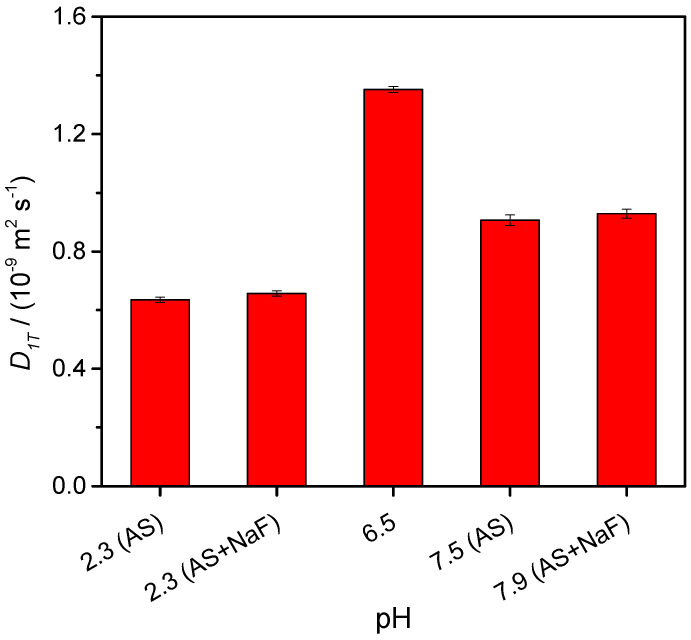
Tracer diffusion coefficients, *D*_T_*,* of NH_4_VO_3_ in saliva artificial (AS) at different pH values_,_ at 25.00 °C. NaF represents sodium fluoride; pH = 6.5 is the value reached from the solutions obtained by dissolving NH_4_VO_3_ in water.

Table 2 and Table 3 show our results of ternary diffusion coefficients for aqueous systems containing NH_4_VO_3_ (*C*_1_) and different carriers (β-CD, HPβ-CD, NaHy, and SDS) (*C*_2_).

**Table 2 biomolecules-12-00947-t002:** Experimental ternary diffusion coefficients (*D*_11_, *D*_12_, *D*_21_, and *D*_22_) of aqueous NH_4_VO_3_ (*C*_1_) + CDs (*C*_2_) solutions and at *T* = 25.00 °C and *P* = 101.3 kPa.

*C* _1_ ^a^	*C* _2_ ^a^	*X* _1_ ^b^	*D*_11_ ± *S*_D_ ^c^	*D*_12_ ± *S*_D_ ^c^	*D*_21_ ± *S*_D_ ^c^	*D*_22_ ± *S*_D_ ^c^
NH_4_VO_3_ (component 1) + β-CD (component 2)						
0.000	0.001	0.000	1.380 ± 0.005	−0.007 ± 0.001	0.035 ± 0.010	0460 ± 0.010
0.0005	0.0005	0.500	1.280 ± 0.010	−0.200 ± 0.040	0.050 ± 0.010	0.400 ± 0.015
0.001	0.000	1.000	0.999 ± 0.010	−0.550 ± 0.020	0.030 ± 0.010	0.380 ± 0.010
NH_4_VO_3_ (component 1) + HP-β-CD (component 2)						
0.000	0.001	0.000	1.200 ± 0.020	−0.012 ± 0.001	0.017 ± 0.005	0439 ± 0.010
0.0005	0.0005	0.500	1.260 ± 0.010	−0.017 ± 0.040	0.013 ± 0.010	0.438 ± 0.015
0.001	0.000	1.000	1.326 ± 0.010	−0.073 ± 0.010	−0.002 ± 0.009	0.437 ± 0.015

^a ^*C*_1_ and *C*_2_ in units of (mol dm^−3^). ^b^
*X*_1_ = *C*_1_/(*C*_1_ + *C*_2_) represents the NH_4_VO_3_ solute mole fraction. ^c^ (*D*_ij_ ± *S*_Dij_) represents the average diffusion coefficients from 6 to 8 replicate measurements and the respective standard deviation in units of (10^9^ m^2^ s^−1^).

Average values of *D*_11_, *D*_12_, *D*_21_, and *D*_22_ were obtained from at least six replicate measurements for each {NH_4_VO_3_ (1) + β-CD (or HP-β-CD or NaHy or SDS (2)} carrier solution composition. Along with each average value, its standard deviation relative to that mean is also presented. *D*_11_ and *D*_22_ were generally reproducible within (±0.02 × 10^−9^ m^2^ s^−1^), while the cross-coefficients *D*_12_ and *D*_21_ were reproducible within (±0.05 × 10^−9^ m^2^ s^−1^). It should be noted that, due to the high viscosity of NaHy, it was not possible to measure these parameters in other compositions (*X*_1_ = 0).

## 4. Discussion

### 4.1. Tracer Diffusion Coefficients of NH_4_VO_3_ in Artificial Saliva at Different pH Values

By analysis of Figure 1, it was verified that there was an accentuated decrease in the tracer diffusion coefficients of NH_4_VO_3_ in all media, at most 53%, when compared with those obtained in water. The decrease in these *D*^0^ values when compared with the *D*^0^ value in water indicated the presence of salting-in effects for NH_4_VO_3_. These vanadate anions (VO_3_^−^) that consist of a complex mixture of different oligomers with different states of protonation [54] suffer more frictional resistance to motion through the fluid, and consequently, their diffusion coefficients in these media become lower, and they can remain retained in the oral cavity, which can cause severe disturbances associated with the potential toxicity of those ions. These salting effects, which are more relevant in artificial saliva, either in the absence or in the presence of sodium fluoride, and at a low pH value (i.e., pH 2.3), can be interpreted on the basis of an electrostatic mechanism; that is, keeping in mind that, in acid solutions, H^+^ is one of the predominant species, due to its large mobility, a strong electric field is generated by a concentration gradient in H^+^. Slowing down these H^+^ ions drive large counter-current fluxes of NH_4_^+^ in aqueous solutions, and consequently, their values of *D* < 0 (salting-in effect).

In contrast, the tracer diffusion coefficients of NH_4_VO_3_ in artificial saliva, with and without NaF at two pH values (7.5 and 7.9), revealed very close values to each other, the differences of which were almost zero, falling within the imprecision margin of this method (<2%).

### 4.2. Ternary and Pseudo Diffusion Coefficients of Aqueous Systems Containing NH_4_VO_3_ and Different Carriers

In Table 2 and Table 3, experimental ternary diffusion coefficients (*D*_11_, *D*_12_, *D*_21_, and *D*_22_) of aqueous NH_4_VO_3_(*C*_1_) plus different components (i.e., CDs, NaHy, and SDS) (*C*_2_) are listed.

From the analysis of these values, it was revealed that, for all systems, the cross-coefficient *D*_21_ values were practically zero, within the uncertainty limits of the measurements, and thus, the influence of NH_4_VO_3_ on the transport of these components (i.e., β-CD, HPβ-CD, SDS and NaHy) was practically null. This fact can be interpreted if we consider the similarity of the mobilities of the free species (β-CD, HP-β-CD, SDS, or NaHy) and the eventual aggregates of these species and NH_4_VO_3_ [55,56].

However, contrary to the *D*_21_ values, i.e., the cross-coefficient *D*_12_ values for some systems were not negligible within the experimental error; that is, coupled diffusion of NH_4_VO_3_ and β-CD occurred, as indicated by non-zero values of the cross-diffusion coefficients, *D*_12_ < 0. β-CD concentration gradients produced significant counter-current coupled flows of NH_4_VO_3_. A possible explanation for these observations is the presence of binding interactions between NH_4_VO_3_ and β-CD molecules, a fact supported by NMR data in our previous study [54] and other studies [54].

In contrast, for aqueous {NH_4_VO_3_ (*C*_1_)/ NaHy (*C*_2_)} system, we observed *D*_12_ > 0. These results revealed that the NaHy concentration gradients could drive significant coupled flows of NH_4_VO_3_, consequently leading to unfavorable conditions for the formation of inclusion complexes with this sterically hindered carbohydrate in solution.

Relative to the other systems involving HP-β-CD, we verified that *D*_12_ = 0. These diffusion data showed that the macromolecular cyclodextrin did not influence the diffusion of the NH_4_VO_3_ component and, under these circumstances, suggested that there was indeed no interaction between HP-β-CD and NH_4_VO_3_. The effect of HP-β-CD on the motion of this salt may be associated with the obstruction that these large molecules exerted on the motion of the small one.

Regarding the other ternary system {NH_4_VO_3_ (*C*_1_)/SDS (*C*_2_)}, also from *D*_12_ = 0, we may conclude that the effect of SDS on the transport of this salt was not accentuated, and consequently, we could infer that the interactions between these solutes (NH_4_VO_3_ and SDS) were almost negligible. These facts were not surprising, considering that the ternary diffusion values for these particular ternary systems were measured at pre-micellar concentrations of SDS; thus, the favorable conditions for having some association between this salt and micelles were not absent [57].

However, in all ternary systems, the limit *D*_12_ → 0, as *X*_1_ → 0, because NaHy (or β-CD, HP-β-CD, and SDS) concentration gradients cannot drive coupled flows of NH_4_VO_3_ in solutions that do not contain NH_4_VO_3_.

By using the ternary diffusion coefficient ratios *D*_21_/*D*_11_ and *D*_12_/*D*_22_, information about coupled diffusion was also obtained. The calculated values were useful once they provided the number of moles of each component transported per mole of the other component (Table 4).

From this table, the higher negative value obtained for *D*_12_/*D*_22_ in the NH_4_VO_3_/β-CD system, when compared with the others, stands out; therefore, we can say that one mole of diffusing β-CD counter-transports up to 1.44 mol of NH_4_VO_3_. From these observations, we can conclude that, among these studied pharmacological compounds, the best for the elimination of vanadium ions eventually present in the oral cavity is β-CD.

## 5. Conclusions

The present study showed that the diffusion coefficients of ammonium vanadate decreased in all aqueous media. These observations were indicative of interactions between the vanadate ions and the other predominant species present (salting-in effects). These ions suffer more frictional resistance to motion through the fluid, and consequently, their diffusion coefficients in these media become lower and can flow slower inside living tissues, causing severe disturbances associated with these ions.

Furthermore, other valuable information was also obtained in order to eliminate, from the oral cavity, the vanadium element resulting from the tribocorrosion to which Ti-6Al-4V prosthetic devices are subject. Our findings revealed that β-CD is the pharmacological compound capable of interacting with more concentrations of vanadium ions.

Although further studies are needed, our results suggest that rinsing with mouthwashes containing β-CDs can be useful in the elimination of vanadium from the oral cavity, with the final purpose of achieving oral prosthetic rehabilitations with greater longevity and maintaining the systemic health of our patients.

## Figures and Tables

**Table 1 biomolecules-12-00947-t001:** Sample description.

Chemical Name	Source	CAS Number	Mass Fraction Purity
NH_4_VO_3_	Merck	7803-55-6	≥0.99 ^a^
NaF	Sigma-Aldrich	7681-49-4	>0.99 ^a^
Lactic acid	Sigma-Aldrich	50-21-5	>0.85 wt% ^a^
Artificial saliva ^b^			
β-CD	Sigma-Aldrich (Water mass fraction of 0.131) ^c^	7585-39-9	>0.97
2-Hydroxypropyl-β-cyclodextrin (HP-β-CD)	(Water mass fraction of 0.03) ^d^	128446-35-5	>0.97
Sodium dodecyl sulfate (SDS)	Merck	7732-18-5	>0.99
NaHy			
H_2_O	Millipore-Q water (1.82 × 10^5^ Ω m at 25.00 °C)	7732-18-5	

^a^ As stated by the supplier. ^b^ Artificial saliva was prepared according the following composition [45,46]: potassium chloride (KCl): 20 mmol/L; sodium bicarbonate (NaHCO_3_): 17.9 mmol/L, sodium phosphate (NaH_2_PO_4_°H_2_O): 3.6 mmol/L, potassium thiocyanate (KSCN): 5.1 mmol/L and lactic acid: 0.10 mmol/L. ^c^ The mass fraction purity is on water-free basis; these data are provided by the suppliers. ^d^ The water content was determined by Karl Fischer method in our laboratory, and the corresponding value obtained was taken into account to determine the solution concentration.

**Table 3 biomolecules-12-00947-t003:** Experimental apparent ternary diffusion coefficients (*D*_11_, *D*_12_, *D*_21_, and *D*_22_) of aqueous NH_4_VO_3_ (*C*_1_) + two carriers (NaHy or SDS, at *C*_2_ = 0) solutions and at *T* = 25.00 °C and *P* = 101.3 kPa.

*C* _1_ ^a^	*C* _2_ ^a^	*X* _1_ ^b^	*D*_11_ ± *S*_D_ ^c^	*D*_12_ ± *S*_D_ ^c^	*D*_21_ ± *S*_D_ ^c^	*D*_22_ ± *S*_D_ ^c^
NH_4_VO_3_ (component 1) + NaHy ^d^ (component 2)						
0.0005	0.0005	0.500	1.360 ± 0.014	0.106 ± 0.010	−0.002 ± 0.008	0.390 ± 0.001
0.001	0.000	1.000	1.352 ± 0.003	0.402 ± 0.090	−0.004 ± 0.007	0.386 ± 0.056
NH_4_VO_3_ (component 1) + SDS (component 2)						
0.001	0.000	1.000	1.334 ± 0.013	0.007 ± 0.010	−0.005 ± 0.008	0.875 ± 0.001
0.0005	0.0005	0.500	1.328 ± 0.014	0.006 ± 0.010	−0.007 ± 0.008	0.899 ± 0.002
0.000	0.001	0.000	1.324 ± 0.010	0.002 ± 0.001	0.085 ± 0.008	0.920 ± 0.001

^a^*C*_1_ and *C*_2_ in units of (mol dm^−3^). ^b^
*X*_1_ = *C*_1_/(*C*_1_ + *C*_2_) represents the NH_4_VO_3_ solute mole fraction. ^c^ (*D*_ij_ ± *S*_Dij_) represents the average diffusion coefficients from 6 to 8 replicate measurements and the respective standard deviation in units of (10^9^ m^2^ s^−1^). ^d^ It should be noted that, due to the high viscosity of NaHy, it was not possible to measure these parameters in other compositions (*X*_1_ = 0).

**Table 4 biomolecules-12-00947-t004:** Moles transporting NH_4_VO_3_ (*C*_1_ = 0.001 mol dm^−3^) by different potential carriers (β-CD, HP-β-CD, NaHy, and SDS).

Aqueous Systems	*D*_12_/*D*_22_^a^
NH_4_VO_3_/β-CD	−1.447
NH_4_VO_3_/HP β-CD	−0.167
NH_4_VO_3_/NaHy	+1.041
NH_4_VO_3_3/SDS	+0.008

^a^*D*_12_ and *D*_22_ values represent the cross-diffusion coefficients, shown in Table 3 and Table 4.

## Data Availability

Data are contained within the article.

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
