# Peer review of "Diffusion of Vanadium Ions in Artificial Saliva and Its Elimination from the Oral Cavity by Pharmacological Compounds Present in Mouthwashes"

_biomolecules, 2022, doi:10.3390/biom12070947_

Round 1
Reviewer 1 Report
This is a very interesting manuscript dealing with the determination of diffusion coefficients of vanadium compounds alone or in combination in artificial saliva intended to simulate the eliminating process in patients with dental prothesis involving this metal in composition. As I can see the experiments and calculations were well performed and the manuscript could be accepted after checking several minor typing point as indicated in the attached pdf file. In particular the following:
1. Please check the redaction because there are some sentences in futre instead of past if considering that the study was carried out.
2. Line 86: Please indicate that dissolving effect is possible if SDS concentration is below the respective CMC. If micelar region is considered the effect is solubilization instead of dissolution.
3. There are several typo mistake as indicated that must be checked.
3.

Author Response
Manuscript Number: biomolecules-1784851
Title: Diffusion of vanadium ions in artificial saliva and its elimination from the oral cavity by pharmacological compounds present in mouthwashes
Authors: Sónia I.G. Fangaia, Ana M.T.D.P.V. Cabral, Pedro M.G. Nicolau, Fernando A.D. R.A. Guerra, M. Melia Rodrigo, Ana C.F. Ribeiro, Artur J.M. Valente, Miguel A. Esteso
Reviewer 1:
This is a very interesting manuscript dealing with the determination of diffusion coefficients of vanadium compounds alone or in combination in artificial saliva intended to simulate the eliminating process in patients with dental prothesis involving this metal in composition. As I can see the experiments and calculations were well performed and the manuscript could be accepted after checking several minor typing point as indicated in the attached pdf file. In particular the following:
We are grateful for these positive comments.
- Please check the redaction because there are some sentences in future instead of past if considering that the study was carried out. The referee is right and, consequently, we have modified the text accordingly.
- Line 86: Please indicate that dissolving effect is possible if SDS concentration is below the respective CMC. If micelar region is considered the effect is solubilization instead of dissolution.
The referee is right and, consequently, we have modified the text accordingly.
- There are several typo mistake as indicated that must be checked.
We really sorry for these mistakes. Consequently, we have checked the text accordingly.
Thanks
Reviewer 2 Report
Review – manuscript no. biomolecules-1784851
Review of the manuscript which has been submitted to Biomolecules
Manuscript no. biomolecules-1784851
Title: Diffusion of vanadium ions in artificial saliva and its elimination from the oral cavity by pharmacological compounds present in mouthwashes
In the current context of the study topic, the article entitled “Diffusion of vanadium ions in artificial saliva and its elimination from the oral cavity by pharmacological compounds present in mouthwashes” is well chosen and the scientific relevance is high considering the increase in the number of dental implants worldwide. Anyway, below I have some questions, observations and I made some suggestions to improve the quality of the work in order to recommend acceptance for publication.
· I can't comment on the scientific resonance of “Taylor dispersion method” for analysis of these systems, but I believe that the authors can supplement the article with more methods in order to support their hypotheses. I think, however, that the use of only 2 methods of analysis is not enough for the publication of an article. The pH measurements were made to prepare the solutions or to monitor the results? It will be interesting to see some variations versus different pH variations-maybe a graphic.
· Also, the presence of some graphics could clarify the content a little. It is difficult to trace the scientific narration between formulas, writing and tables.
· it would be interesting to see a graph with the variation of the diffusion coefficients depending on the diffusion medium.
· Please explain clearly in the manuscript what a diffusion coefficient of 0 means and what it means to be higher as a practical application in the present study
Page 1, row 18: please reformulate “the form ammonium salt”;
Page 1, row 26: please reformulate “the form ammonium salt”;
Page 1, rows 35-37: please reformulate for a better understanding “Titanium-aluminium-vanadium alloys are commonly used in the fabrication of dental implants, and their prosthetic supra-structures, abutments and healing components, and also orthopedic implants, for having good mechanical properties and biocompatibility”
Page 3, row 98: I don’t understand the section “2. Isothermal diffusion of ammonium vanadate in multicomponent systems: Definition of this phenomenon and highlighting of inherent concepts” presence in the article. The authors will have to efficiently distribute the information from Section 2 among the other sections, more exactly the Introduction, Materials and Methods, Results and Discussion Sections.
Page 4-6, rows 163-229: Certain paragraphs of the description of the experimental method can be included in the Results or Discussions text body. The purpose of the Materials and Methods section is to describe the method, not the method principle or why it was chosen, etc. as it is described now.
Page 6, rows 240-244: please reformulate for a better understanding: “Table 2 present our data of the diffusion coefficients for NH4VO3 at tracer concentrations in different media. That is, water, artificial saliva without and with other components (lactic acid, sodium fluoride, and the mixture of both), at different pHs, measured by Taylor equipment at 25.00 °C and tracer concentrations, once that in practice that the concentration of these ions resulting from the corrosion of the dental alloys is very small”;
Page 6, row 245: you missed a bibliographic reference;
Page 6, row 249: please reformulate for a better understanding “in saliva artificial at different”;
Page 6, row 252: please reformulate for a better understanding “and without and with NaF”
Page 6, row 254: please reformulate for a better understanding “aqueous solutions NH4VO3 0.001 mol dm-3”;
Page 7, row 258: How do the readers know which results are for each of “β-CD, HPβ-CD, NaHy and SDS” systems? Please change for a clearer understanding. The same observation for the Table 4. The authors must clarify the tables for a better understanding.
Page 9, row 345-347: please reformulate more clearly the sentence for a better understanding “The present study showed that the diffusion coefficients of vanadium ions in the form ammonium salt, decreased in all aqueous media, which may indicate that these ions remain longer in the areas where they are released by prosthetic devices.”
Page 8, row 309: why did the authors not add this data in the article “fact supported from our NMR data”
Author Response
Manuscript Number: biomolecules-1784851
Title: Diffusion of vanadium ions in artificial saliva and its elimination from the oral cavity by pharmacological compounds present in mouthwashes
Authors: Sónia I.G. Fangaia, Ana M.T.D.P.V. Cabral, Pedro M.G. Nicolau, Fernando A.D. R.A. Guerra, M. Melia Rodrigo, Ana C.F. Ribeiro, Artur J.M. Valente, Miguel A. Esteso
Reviewer 2:
In the current context of the study topic, the article entitled “Diffusion of vanadium ions in artificial saliva and its elimination from the oral cavity by pharmacological compounds present in mouthwashes” is well chosen and the scientific relevance is high considering the increase in the number of dental implants worldwide. Anyway, below I have some questions, observations and I made some suggestions to improve the quality of the work in order to recommend acceptance for publication.
We are grateful for these positive comments.
1) I can't comment on the scientific resonance of “Taylor dispersion method” for analysis of these systems, but I believe that the authors can supplement the article with more methods in order to support their hypotheses. I think, however, that the use of only 2 methods of analysis is not enough for the publication of an article. The pH measurements were made to prepare the solutions or to monitor the results? It will be interesting to see some variations versus different pH variations-maybe a graphic.
We are grateful for these valuable comments. In fact, in this work, we have used the Taylor dispersion method for the measurement of diffusion coefficients in aqueous systems containing ammonium vanadate, considered ternary ones, at different concentrations and different pH values. In addition, the pH measurements were carried out to monitor the results. That is, to help understand the behaviour of diffusion of these ions in different media (acid, basic and neutral).
Although we understand the reviewer’s point of view, we must highlight that the multicomponent diffusion measurements involve experimental and mathematical complexity, which justify the scarcity of available information and researchers using this technique as well.
Our main goal is to contribute for a deeper understanding of the fundamental diffusion properties of these systems in order to improve their efficacy in clinical therapeutics. That is, the diffusion of such vanadium ions into the organism, as a consequence of the tribocorrosion of Ti-6Al-4V prosthetic devices, and carried by saliva, can cause health problems as a consequence of their toxicity, enhanced by a cumulative effect in the body. The observed effect of cyclodextrins (commonly used in mouthwash formulations) on vanadium ions diffusion allowed us to conclude that there is a strong interaction between these two components, considering, thus, that the presence of these macrocycles in the oral cavity is very useful in the elimination of the vanadium ions. These transport properties are thus of great interest not only for fundamental purposes but also for this technical and biomedical applications.
Based on your suggestion, we have changed the Table 2 by a Figure with diffusion coefficients as a function of pH values.
2)Also, the presence of some graphics could clarify the content a little. It is difficult to trace the scientific narration between formulas, writing and tables. it would be interesting to see a graph with the variation of the diffusion coefficients depending on the diffusion medium.
This question is already answered in point 1. Concerning, other tables, from our experience that is not easy given the data complexicity.
3) Please explain clearly in the manuscript what a diffusion coefficient of 0 means and what it means to be higher as a practical application in the present study
We are grateful for these comments. In fact, the referee is right relative to the meaning of the diffusion coefficient of NH4VO3 of 0 because it is not very clear in the text. There are some concepts not trivial in this manuscript. Firstly, we can say that these parameters measured in those circumstances (i.e., indicated as D0) were obtained at an infinitesimal concentration (or infinite dilution). For that, the dispersion profiles were prepared by injecting small volumes (approximately 0.063 cm3) of different solutions of NH4VO3 at different concentrations (i.e., 0.001, 0.002, 0.005, 0.008 and 0.010 mol dm-3) which will be introduced later into a laminar stream (that is, in this case, water). In addition, we can say also those parameters are obtained at tracer concentrations because this salt only is present in the injected solutions and not in flow solutions.
Diffusion coefficients, DT, of NH4VO3 in other media also were measured (e.g., saliva artificial at different pHs) at tracer concentrations, because this salt is also present in the injected solutions. The only difference is the flow solution (now, it is artificial saliva, not water). From these observations, and having in mind a better clarification, we adopt the criteria of calling tracer diffusion coefficients of NH4VO3 to all parameters measured at tracer concentrations (knowing also as limiting diffusion coefficients), independently of the carrier solution and ignore the index 0. In addition, we have also deleted the index 1 to indicate component 1 in these systems. In the other words, in Table 2, we have represented the diffusion coefficients of NH4VO3 at tracer concentrations in all media by DT instead D01T.
4) Page 1, row 18: please reformulate “the form ammonium salt”;
The text has been changed to “ammonium vanadate”.
5) Page 1 row 26: please reformulate “the form ammonium salt”;
The text has been changed to “ammonium vanadate”.
6) Page 1, rows 35-37: please reformulate for a better understanding “Titanium-aluminium-vanadium alloys are commonly used in the fabrication of dental implants, and their prosthetic supra-structures, abutments and healing components, and also orthopedic implants, for having good mechanical properties and biocompatibility”
This statement has been simplified and now reads: “Titanium-aluminium-vanadium alloys are commonly used in the fabrication of orthopedic and dental implants, as well as in dental prosthetic supra-structures, abutments and healing components, due to their good mechanical properties and biocompatibility [1–3]
7) Page 3, row 98: I don’t understand the section “2. Isothermal diffusion of ammonium vanadate in multicomponent systems: Definition of this phenomenon and highlighting of inherent concepts” presence in the article. The authors will have to efficiently distribute the information from Section 2 among the other sections, more exactly the Introduction, Materials and Methods, Results and Discussion Sections.
We are grateful for these positive comments. We agree with the referee relative to distribution of the information from Section 2 among the other sections, and, consequently, we have modified the text, accordingly.
8) Page 4-6, rows 163-229: Certain paragraphs of the description of the experimental method can be included in the Results or Discussions text body. The purpose of the Materials and Methods section is to describe the method, not the method principle or why it was chosen, etc. as it is described now.
We really appreciated these comments but as it can be inferred from the previous reviewer’s comments, these measurements are not trivial as well as the inherent concepts. Therefore we feel that this description should be kept as is.
9)Page 6, rows 240-244: please reformulate for a better understanding: “Table 2 present our data of the diffusion coefficients for NH4VO3 at tracer concentrations in different media. That is, water, artificial saliva without and with other components (lactic acid, sodium fluoride, and the mixture of both), at different pHs, measured by Taylor equipment at 25.00 °C and tracer concentrations, once that in practice that the concentration of these ions resulting from the corrosion of the dental alloys is very small”;
The text has been modified by following the reviewer’s comment.
10) Page 6, row 245: you missed a bibliographic reference;
We really sorry for the missing a a bibliographic reference. The referee is right and, consequently, we have inserted the text accordingly.
11) Page 6, row 249: please reformulate for a better understanding “in saliva artificial at different”;
The referee is right and, consequently, we have modified the text accordingly.
12) Page 6, row 252: please reformulate for a better understanding “and without and with NaF”
The text has been reformulated accordingly.
13) Page 6, row 254: please reformulate for a better understanding “aqueous solutions NH4VO3 0.001 mol dm-3”;
This has be done.
14) Page 7, row 258: How do the readers know which results are for each of “β-CD, HPβ-CD, NaHy and SDS” systems? Please change for a clearer understanding. The same observation for the Table 4. The authors must clarify the tables for a better understanding.
To change C1 and C2 by the concentrations of specific compounds will increase the number of tables. As it is, we can just indicate what species are realted with components 1 and 2. In any case, we found that, in few cases, the desciprion of component (1) and (2) were missing; in the revised version this was updated.
15) Page 9, row 345-347: please reformulate more clearly the sentence for a better understanding “The present study showed that the diffusion coefficients of vanadium ions in the form ammonium salt, decreased in all aqueous media, which may indicate that these ions remain longer in the areas where they are released by prosthetic devices.”
The referee is right and, consequently, we have modified the text accordingly.
16) Page 8, row 309: why did the authors not add this data in the article “fact supported from our NMR data”
Our apologies if the sentence is not clear. In fact, the association between ammonium monovanadate and β-Cyclodextrin was previously evaluated by us using NMR and transport techniques, and computational calculations. To avoid self-plagiarism a citation to ref 55 was displaced for a better understanding.
Round 2
Reviewer 2 Report
The authors made all the suggested changes and even more. Consequently, the manuscript is easy to read and understand due to the clear explanations.
Thanks!